# Physicochemical Antioxidative and Emulsifying Properties of Soybean Protein Hydrolysates Obtained with Dissimilar Hybrid Nanoflowers

**DOI:** 10.3390/foods11213409

**Published:** 2022-10-28

**Authors:** Geng Li, Jingwen Xu, Huiwen Wang, Lianzhou Jiang, Huan Wang, Yan Zhang, Hua Jin, Zhijun Fan, Jing Xu, Qingshan Zhao

**Affiliations:** 1College of Arts and Sciences, Northeast Agricultural University, Harbin 150030, China; 2College of Food Science, Northeast Agricultural University, Harbin 150030, China; 3Coastal Research and Extension Center, Mississippi State University, Starkville, MS 39762, USA; 4Heilongjiang Beidahuang Green and Healthy Food Co., Ltd., Jiamusi 154007, China; 5Experimental Practice and Demonstration Center, Northeast Agricultural University, Harbin 150030, China

**Keywords:** hybrid nanoflowers, soybean protein, hydrolysates, structure properties, functional properties, emulsion

## Abstract

This study investigated the changes in the structure and properties of soybean protein after hydrolysis using two types of hybrid nanoflowers (alcalase@Cu_3_(PO_4_)_2_•3H_2_O (ACHNs) and dispase@Cu_3_(PO_4_)_2_•3H_2_O (DCHNs)) and examined the basic properties and oxidative stability of hydrolyzed soybean protein emulsions. The formations of the two hybrid nanoflowers were first determined using a scanning electron microscope, transmission electron microscope, and Fourier infrared spectroscopy. The structure and functional properties of soybean protein treated with hybrid nanoflowers were then characterized. The results indicated that the degree of hydrolysis (DH) of the ACHNs hydrolysates was higher than that of the DCHNs for an identical reaction time. Soybean protein hydrolysates treated with two hybrid nanoflowers showed different fluorescence and circular dichroism spectra. The solubility of the hydrolysates was significantly higher (*p* < 0.05) than that of the soybean protein (SPI) at all pH values tested (2.0–10.0)*: at the same pH value, the maximum solubility of ACHNs hydrolysates and DCHNs hydrolysates was increased by 46.2% and 42.2%, respectively. In addition, the ACHNs hydrolysates showed the highest antioxidant activity (DPPH IC_50_ = 0.553 ± 0.009 mg/mL, ABTS IC_50_ = 0.219 ± 0.019 mg/mL, and Fe^2+^ chelating activity IC_50_ = 40.947 ± 3.685 μg/mL). The emulsifying activity index of ACHNs and DCHNs hydrolysates reached its maximum after hydrolysis for 120 min at 61.38 ± 0.025 m^2^/g and 54.73 ± 0.75 m^2^/g, respectively. It was concluded that the two hydrolysates have better solubility and antioxidant properties, which provides a theoretical basis for SPI product development. More importantly, the basic properties and oxidative stability of the soybean-protein-hydrolysates oil-in-water emulsions were improved. These results show the importance of proteins hydrolyzed by hybrid nanoflowers as emulsifiers and antioxidants in the food and pharmaceutical industry.

## 1. Introduction

Soybeans are among the most widely planted legumes globally and are important sources of protein, given their high nutritional value. Soybean protein is rich in all essential amino acids, and its nutritional and biological value is comparable to animal proteins; it can replace animal proteins in the food industry [1]. However, the functional properties of natural soybean protein are relatively poor, which greatly limits their application in the food industry [2]. Therefore, broadening the market applications of soybean protein would require its structure to be modified to enhance its functional properties and biological activity. At present, protein modification technologies focus mainly on physical and chemical modification and hydrolysis. Physical modification involves no addition of exogenous substances, but the modification effect is limited; chemical modification is the induction of new organic groups into the protein molecule, which causes changes in the protein main chain or side chains, resulting in changes in the protein structure and physicochemical properties. However, chemical modification also has limitations in its application in the food industry because of the addition of chemicals that may be harmful to human health. Enzymatic modification of proteins is a protein modification means to convert proteins into peptides of different molecular weights by using proteases to catalyze the hydrolysis of peptide bonds in proteins under suitable conditions. Compared with physical and chemical modifications, enzymatic modification has the advantages of less by-product, high specificity and easy control, and it is widely used [3].

Hydrolysis requires mild reaction conditions. The use of extreme environmental conditions during the modification process and production of side reactants should be avoided to maintain the nutritional value of the protein after modification [4,5]. During hydrolysis, the molecular weight of the protein decreases, hydrophobic groups are exposed, and the number of ionized groups increases [6]. The structural changes in soybean protein after hydrolysis give it better functional properties, such as good solubility and better emulsification [7]. Protein hydrolysis results in a different degree of hydrolysis (DH) at different reaction times, and these determine the structure and functional properties of the product [8]. A high DH produces many short-chain peptides that worsen the protein properties. In contrast, a lower DH in hydrophobic and hydrophilic residues will strengthen the amphiphilic properties of the protein and improve its functional properties [9]. In addition, the experimental results of Xu et al. [10] show that the degree of protein hydrolysis also has an effect on the stability of the emulsions prepared from the hydrolysates. 

In addition to its functional properties, the potential biological activities of hydrolyzed protein–such as its antioxidant properties–have also been studied extensively. Zheng et al. [11] studied the free-radical-scavenging ability of black soybean protein hydrolyzed by ficin, bromelain, and alcalase. Similarly, Calderón-Chiu et al. [12] found that hydrolyzing jackfruit leaf protein using pepsin and pancreatin improved its antioxidant activity. In addition, Wang et al. [13] reported that hydrolyzed proteins can maintain the oxidative stability of an unsaturated fatty acid system. Therefore, protein hydrolysates with good functional properties have attracted much attention in nutrition, medicine, and the food industry [14].

The protease in the hybrid nanoflowers, when compared to the free enzyme, combines with inorganic ions to form a flower-like structure [15]. The structure has a large specific surface area, and the effective restriction of enzyme molecules promotes the activity and stability, with improved reusability [16,17]. Recently, topics from the hydrolysis of proteins using hybrid nanoflowers to the functional properties of proteins have attracted widespread attention. Memon et al. [18] studied the structure, morphology, and compositional characteristics of an alcalase@CaHPO_4_ hybrid nanoflower and discussed its solubility, ability to scavenge diphenyl-1-picrylhydrazyl (DPPH) radicals, and the calcium-binding ability of the SPI hydrolysate. Feng et al. [19] synthesized a new type of papain-Cu_3_(PO_4_)_2_•3H_2_O-magnetic nanoflower (PCMN) that can hydrolyze most of the allergen proteins in milk to produce low-sensitivity milk. However, there are few reports about the hydrolyzed protein formed from hybrid nanoflowers using different enzymes and the functional properties of the hydrolysate at different times. This experiment is the first time the structural and functional properties of the hydrolysates of nano-flowers hybrid with alkaline protease and neutral protease are being compared. Therefore, in this study, alcalase and dispase were immobilized with metal Cu^2+^, and the formation of hybrid nanoflowers was determined using electron microscopy and infrared spectroscopy. The structures and functional properties of soybean protein hydrolysates under different DHs were studied. Thus, it is helpful to explore the application potential and promote the research and development of healthy, functional food. At the same time, based on their emulsifying properties and antioxidant capacity, we prepared oil-in-water emulsions to evaluate their basic properties and ability to inhibit lipid oxidation. Exploring the use of protein hydrolysates as potential functional components, this laid a theoretical foundation for the development of protein hydrolysates obtained using hybrid nanoflowers with the excellent performance required for application in the medicinal and food industries.

## 2. Materials and Methods

### 2.1. Materials

Defatted soybean flour was from Shandong Zhaoyuan Food Co., Ltd. (Shandong, China). Alcalase was purchased from Novozymes Biotechnology Co., Ltd. (Tianjin, China). Dispase was offered by Beijing Boao Tuoda Technology Co., Ltd. (Beijing, China). Soybean oil came from a shop in Harbin, Heilongjiang, China. 2,2-diphenyl-1-(2,4,6-trinitrophenyl)-hydrazyl (DPPH), 2,2′-Azinobis-(3-ethylbenzthiazoline-6-sulphonate) (ABTS), ferrozine, and SDS were provided by Shanghai Aladdin Biochemical Technology Co., Ltd. (Shanghai, China). The remaining chemical reagents of the experiment were of analytical purity grade.

### 2.2. Preparation of Hybrid Nanoflowers

The wet-chemical precipitation approach described by Memon et al. [18] was used, with some modification, to prepare the alcalase@Cu_3_(PO_4_)_2_•3H_2_O hybrid nanoflowers (ACHNs) and dispase@Cu_3_(PO_4_)_2_•3H_2_O hybrid nanoflowers (DCHNs). Briefly, 100 μL alcalase/dispase (50 mg/mL) was dissolved in 4.5 mL phosphate-buffered saline (PBS; 10 mM, pH 7.4), and 100 μL of CuCl_2_ (180 mM) solution was introduced into the former and stirred using a magnetic stirrer. The solution was blended at 25 °C for 48 h. In addition, the precipitate was collected and freeze-dried (LYOQUEST-85PLUS, Telstar, Spain) for later use.

### 2.3. Characterization of Hybrid Nanoflowers

The nanoflower lyophilized samples were gold-sprayed to coat the surface with gold. Then, observation was performed using a scanning electron microscope (SEM) (SU8010, Hitachi, Japan) at an accelerated voltage of 5.0 KV.

The microstructure of the nanoflower samples was observed using H-7650 transmission electron microscope (TEM) (H-7650, Hitachi, Japan). The nanoflowers were diluted to a certain multiple and the diluted nanoflower samples were dropped onto a carbon-coated copper grid and adsorbed for 1 h. Then, the samples were stained with 2% (*w*/*v*) phosphotungstic-acid negative staining solution for 2 min. After the samples were air-dried at room temperature, the nanoflower samples were observed at an accelerating voltage of 100 kV.

In conjunction with Fourier transform infrared (FTIR) spectroscopy (Bruker Vertex 70, Bruker Optics, Ettlingen, Germany), the nanoflower samples were mixed with potassium bromide (1:100, *w*/*w*) and pressed into tablets for processing. The infrared spectra were scanned in the wavelength range of 4000–400 cm^−1^ at a resolution of 4 cm^−1^ and a scanning frequency of 64 times [20]. 

### 2.4. Mass Percentage of Alcalase/Dispase in Hybrid Nanoflowers

To accurately estimate the amount of enzyme added during the hydrolysis of soybean protein using hybrid nanoflowers, it was necessary to determine the percentage of alcalase/dispase in the latter. The muffle furnace (SX_2_-2.5-12, Tianjin Zhonghuan Furnace Corp., Tianjin, China) was preheated at 200–600 °C for 4 h and the dried hybrid nanoflowers were calcinated at 700 °C for 2 h. A precipitate, the inorganic metal salt Cu_3_(PO_4_)_2_, was obtained after water and organic enzyme crystals were removed from the hybrid nanoflowers. The equation used for calculating the mass percentage of the enzymes in the nanoflowers is as follows [21]:(1)$$W\left(\%\right)=\frac{G-G_{i}}{G}\times 100$$
where W is the mass percentage of enzymes in the hybrid nanoflowers (%), G is the mass of the hybrid nanoflowers (g), and G_i_ is the mass of Cu_3_(PO_4_)_2_•3H_2_O(g).

Experimental results show that the mass percentage of the ACHNs and DCHNs are 32.25 ± 0.926% and 26.23 ± 0.71%, respectively.

### 2.5. Preparation of Soybean Protein (SPI)

The SPI was extracted using the practice reported by Zheng et al. [22], with some modifications. Degreased soybean flour was mixed with ultrapure water at 1:10 (w:v), and it was blended at 25 °C for 2 h, with the pH regulated to 8.0. Subsequently, the solution received centrifugation treatment at 4000 rpm for 15 min (SC-3610, Anhui USTC Zonkia Scientific Instruments Co., Ltd., Anhui, China). The supernatant was gathered, with its pH adjusted to 4.5. After being left to incubate overnight, the protein precipitate was gathered and centrifugated at 4000 rpm for 20 min. The precipitate was dissolved in ultrapure water, with solution pH regulated to pH 7.0. The SPI obtained was freeze-dried for later use. The protein content (90.31% ± 0.59%) was determined using the Kjeldahl method.

### 2.6. Hydrolysis of SPI Using Alcalase@Cu_3_(PO_4_)_2_•3H_2_O (ACHNs) and Dispase@Cu_3_(PO_4_)_2_•3H_2_O (DCHNs)

Hydrolysis of SPI was performed, with slight modifications, following the practice of Liu et al. [23]. The SPI (2%, *w*/*v*) was dissolved in ultrapure water and blended at 25 °C for 2 h until fully hydrated. The dispersion was incubated in a water bath, with its temperature adjusted to 50 °C. The solution pH was regulated to the optimum for the enzymes (8.0 for ACHNs, 7.0 for DCHNs) using 0.1 M NaOH. The hybrid nanoflowers were introduced into the solution, based on an enzyme-substrate ratio of 1:100 (*w*/*w*), and 0.1 M NaOH was introduced during the reaction for keeping the system pH consistent at the optimum for the enzymes. The hydrolyzed products were collected after 30, 60, 90, 120, and 180 min, respectively. The reaction was concluded through putting the solution in the boiling water bath for 10 min. Upon cooling, the mixture received centrifugation at 10,000 rpm for 10 min (GL-20G-Ⅱ, Shanghai Anting Scientific Instrument Factory, Shanghai, China) to remove inorganic precipitates. The SPHs obtained were freeze-dried.

### 2.7. Degrees of Hydrolysis

The DH in soybean protein hydrolysates (SPHs) were determined using the Adler-Nissen method [24]. The DH was determined using the following formula:(2)$$\mathrm{DH}\left(\%\right)=\frac{B\times N}{\alpha \times \mu_{p}\times h_{\mathrm{tot}}}\times 100\%$$
where B refers to the NaOH volume consumed by hydrolysis (mL), N indicates the NaOH concentration (0.1 M), α refers to the dissociation degree in amino group (0.885 for ACHNs, 0.44 for DCHNs), μ_p_ represents the weight of the protein (g), and h_t__ot_ indicates the number of millimoles in peptide bonds per gram of protein (7.8 mmol/g).

### 2.8. Structural Characterization of SPI and Its Hydrolysates

#### 2.8.1. Fluorescence Spectroscopy

Fluorescence spectroscopy of the protein hydrolysate was made with the approach in Feng et al. [25], with slight modifications. The SPI/SPHs were prepared at a concentration of 0.2 mg/mL and analyzed with fluorescence spectroscopy and a fluorescence instrument (LS55, PerkinElmer Co., Ltd, Waltham, MA, USA) at an excitation wavelength of 290 nm, a scanning range of 300–500 nm, and excitation and emission slits of 5 nm.

#### 2.8.2. Circular Dichroism Spectroscopy

Circular dichroism (CD) spectroscopy (MOS-4SO, Bio-Logic, Seyssinet-Pariset, France) was adopted for analyzing changes in secondary structure of SPI upon hydrolysis, following Yin et al., [26] with slight modifications. The sample solution was 0.15 mg/mL and the CD spectrum was measured between 190 nm and 260 nm. The scanning rate and resolution were 75 nm/min and 0.1 nm, respectively. The data were processed using the software “CDNN”, and the content of each secondary structure in the protein was calculated.

### 2.9. Physiochemical Properties of SPI and Its Hydrolysates

#### Solubility

The solubility was calculated with the measure used by Klompong et al. [27], with minor modifications. The samples (10 mg/mL) were dispersed in ultrapure water and blended at 25 °C for 2 h. The pH was adjusted to 2.0–10.0 using 1 N HCl or 1 N NaOH. The dispersion liquid obtained received centrifugation at 10,000 rpm for 10 min. Protein contents in hydrolysates were calculated with the Lowry method [28], and the total protein content (N × 5.8) in the supernatants was measured by employing the Kjeldahl method. Equation (3) describes the calculation of solubility (%), as below:(3)$$\mathrm{Solution}\left(\%\right)=\frac{\mathrm{Protein}\;\mathrm{content}\;\mathrm{in}\;\mathrm{the}\;\mathrm{supernatant}}{\mathrm{Total}\;\mathrm{protein}\;\mathrm{content}\;\mathrm{in}\;\mathrm{solution}}\times 100\%$$

### 2.10. Functional Properties of SPI and Its Hydrolysates

#### 2.10.1. Emulsifying Properties

Emulsifying properties in SPI/SPHs were measured following Nasir et al. [29], with minor modifications. A sample solution of 12 mL at a concentration of 1 mg/mL was introduced into 4 mL of soybean oil and homogenized for 2 min at 10,000 rpm with the high-speed homogenizer (FJ200-SH, Shanghai Huxi Industry Co., Ltd., Shanghai, China). After homogenization, 40 μL of liquid was immediately aspirated from the bottom of the emulsion and mixed with 5 mL of sodium dodecyl sulfate in deionized distilled water (SDS solution; 0.1 %, *w*/*v*). The mixture was stirred by the vortex mixer for 5 s, with its absorbance (A_0_/A_10_) calculated at a wavelength of 500 nm (T6, Beijing Purkinje General Instrument Co., Ltd., Beijing, China). The emulsifying activity index (EAI) and emulsifying stability index (ESI) were measured with Equations (4) and (5), as below:(4)$$\mathrm{EAI}\left(m^{2}/g\right)=\frac{2\times 2.303\times A_{0}\times N\times 10^{-4}}{\varphi \times c\times 10000}$$
(5)$$\mathrm{ESI}\left(\min\right)=\frac{A_{0}}{A_{0}-A_{10}}\times 10$$
where N is the dilution factor (125), φ refers to the oil fraction (0.25), and c means the protein concentration (1 mg/mL).

#### 2.10.2. Antioxidant Capacity

##### DPPH Radical-Scavenging Activity

To evaluate the antioxidant activity for peptides precisely, the DPPH radical-scavenging activity was assessed with the approach in Gong et al. [30], with minor modifications. At first, 1 mL of SPI or hydrolysates (0.5–2.5 mg/mL) was introduced into 4 mL of 0.1 mmol/L DPPH ethanol solution (95%, *v*/*v*). The reaction continued in darkness at 25 °C for 30 min. Then its absorbance at 517 nm was calculated, with the ethanol solution (95%, *v*/*v*) alone for blank control. The DPPH radical-scavenging rate was measured using Equation (6):(6)$$\mathrm{DPPH}\;\mathrm{radical}\;\mathrm{scavenging}\;\mathrm{activity}\left(\%\right)=\left(1-\frac{A-A_{i}}{A_{j}}\right)\times 100\%$$
where A means the reaction absorbance in the sample with the DPPH ethanol solution, A_i_ represents the absorbance in-sample after being mixed with an ethanol solution, and A_j_ indicates the absorbance of DPPH mixed with ethanol.

##### ABTS Radical-Scavenging Activity

The ABTS radical-scavenging activity for protein hydrolysate was measured following Pellegrini et al. [31], with slight modifications. ABTS and potassium persulphate were added to 0.1 mol/L phosphate buffer (pH 7.4) to create an ABTS solution at a concentration of 7 mmol/L (containing 2.45 mmol/L potassium persulphate) and stored at 25 °C in darkness for 16 h in order to obtain a cationic ABTS solution.

Following this, 30 μL of the sample solution (0.5–2.5 mg/mL) was blended with 3 mL of the diluted cationic ABTS radical solution before the whole was shaken and mixed and the reaction left to occur in darkness at 25 °C for 5 min. The obtained solution was monitored for absorbance at 734 nm. The ABTS radical-scavenging rate was calculated using Equation (7):(7)$$\mathrm{ABTS}\;\mathrm{radical}\;\mathrm{scavengin}\;\mathrm{activity}\left(\%\right)=\left(1-\frac{A_{i}}{A_{0}}\right)\times 100\%$$
where A_0_ represents the ABTS solution absorbance and A_i_ represents the sample absorbance after reaction with the ABTS solution.

##### Metal Ion Chelating Activity

The ferrous ion (Fe^2+^) chelating activity in protein hydrolysates was calculated using the method of Wang et al. [32], with slight modifications. First, 1 mL of the sample solution (50–250 μg/mL) was added to ultrapure water (3.7 mL) with 0.1 mL of FeCl_2_ solution (20 mmol/L); this was held still for 3 min before 0.2 mL of phenelzine (5 mmol/L) was added to it. After being incubated for 10 min, its solution absorbance was monitored at 562 nm. The ferrous chelating capacity was measured by Equation (8):(8)$$\mathrm{Ferrous}\;\mathrm{chelating}\;\mathrm{activity}\left(\%\right)=\left(1-\frac{A_{i}}{A_{0}}\right)\times 100\%$$
where A_i_ means the absorbance for sample solution and A_0_ means the absorbance for sample solution after replacing it with ultrapure water.

##### Determination of IC_50_ Values

The experimental data were fitted to a nonlinear curve using the software SPSS 20.0, and the concentration of hydrolysates (mg/mL) required for a free-radical-scavenging rate of 50% was set as the IC_50_ value [33].

### 2.11. Preparation of Protein Emulsions

Protein emulsions were produced using soybean protein and their hydrolysates were used as emulsifiers with the method of Xu et al. [34], with minor modifications. The soybean protein and its prepared hydrolysates were dissolved in 0.01 mol/L phosphate buffer (pH 7.0) and blended for 2 h at 25 °C. A 20 mL sample solution (20 mg/mL) was added to the soybean oil (3%, *v*/*v*) and shaken magnetically at 25 °C for 10 min. The sample then received homogenization for 4 min at 10,000 rpm with a high-speed homogenization and dispersion instrument to obtain a crude emulsion. The crude emulsion was immediately treated with an ultrasonic cell crusher (JY92-2D, Ningbo Scientz Biotechnology Co., Ltd., Ningbo, China) at 400 W for 15 min (at a temperature of 25 °C, controlled using an ice-water bath) to obtain an oil-in-water protein emulsion.

### 2.12. Characterizing the Emulsion

#### 2.12.1. Particle Size and Zeta (ζ) Potential Measurement

The mean particle size and zeta-potential in emulsion samples were identified at 25 °C with Zetasizer Nano-ZS90 (Malvern Instruments, Worcestershire, UK) [34]. The emulsion samples were diluted 100 times in 0.01 mol/L phosphate buffer (pH 7.0) before measurement to avoid multiple light-scattering effects.

#### 2.12.2. Oxidation Stability

The emulsions were preserved at 25 °C for eight days, following which the formation of peroxide (POV) values for lipid hydroperoxides, thiobarbituric acid reactive substance (TBARS) values for the products of thiobarbituric acid reactions, and the secondary products for lipid oxidation were determined; the oxidative stability of the emulsions was analyzed using the POV and TBARS values as indicators [35].

Following this, 0.2 mL of the emulsion was mixed with 1.5 mL of a mixture consisting of isopropanol and isooctane (1:3, *v*/*v*). This solution underwent centrifugation at 4000 rpm for 30 min. Afterwards, 2.8 mL of methanol and n-butanol (2:1, *v*/*v*), 15 μL of 3.94 mol/L NH_4_SCN solution, and 15 μL of Fe^2+^ solution were added to 0.2 mL of the supernatant. Subsequently, after the mixture was cultured in darkness for 20 min, its absorbance was measured at 510 nm. Sample POV was measured according to the standard curve of isopropylbenzene peroxide.

The next step was to mix 2 mL of the emulsion with 2 mL of a mixture of TCA—TBA—HCl and BHT—ethanol (100:3, *v*/*v*), with the reaction performed in a boiling water bath for 15 min. Following this, the sample was cooled to room temperature, followed by centrifugation at 4000 rpm for 10 min. TBARS values in the samples were calculated using the standard curve for malondialdehyde.

### 2.13. Statistical Analysis

Every test was conducted thrice in parallel and results were indicated by mean ± standard deviations. One-way analysis by ANOVA using SPSS 20.0 (SPSS Inc., Chicago, IL, USA), with a significant difference of *p* < 0.05. Origin V8.1 ( OriginLab Corporation, Northampton, MA, USA), was used for data processing and graphing.

## 3. Results and Discussion

### 3.1. Structural Characteristics of the Hybrid Nanoflowers

To analyze the effect of hybrid nanoflowers on the structure and properties of soybean protein hydrolysates and the formation of emulsions, it was necessary to ensure the formation of hybrid nanoflower composites. Morphological analysis using SEM and TEM showed an irregular crystalline structure for inorganic carriers (Figure 1A,D) while the hybridized nanoflower composites were flower-like and had hierarchical structures (Figure 1B,C,E,F). The flower-like hierarchical structures of the hybridized nanoflowers may have been created by the interaction of enzymes and carriers [18]. Nanoflowers synthesized using different proteases varied in size, with ACHNs having a larger surface-to-volume ratio than DCHNs due to the presence of different amide groups on the surface of the enzyme. The amide groups provide phosphate crystals to bind flower-like structures and help their growth by providing nucleation sites with different geometrical configurations and densities [36]. These results are consistent with the findings of Yu et al. [37].

The infrared spectrum (Figure 2) shows that the inorganic carriers in ACHNs and DCHNs have a strong absorption at 1037 cm^−1^ and that this absorption is a P-O stretching vibration. The bands at 553 cm^−1^ and 561 cm^−1^ belong to the O-P=O bending vibration. The presence of these characteristic peaks indicates the presence of phosphate groups in the hybridized nanoflowers. These characteristic absorption protease peaks were present in ACHNs and DCHNs at 1400–1600 cm^−1^ for -CONH absorption and at 2800–3000 cm^−1^ for CH_2_ and CH_3_ absorption [18]; this demonstrated the presence of free enzymes. In addition, no new absorption peaks or significant peak shifts were observed for the hybrid nanoflowers, indicating that the protease was immobilized after self-assembly.

### 3.2. Degrees of Hydrolysis

The DH is a measure of how well a protein is hydrolyzed; the DH increases as more peptide bonds are broken [27]. Figure 3 shows the curves of soybean protein hydrolyzed by ACHNs and DCHNs for different hydrolysis times. As shown in the figure, the DH tend to increase with increasing reaction time, as consistent with the gradual release of protein peptides during the hydrolysis process [15]. During the hydrolysis of soybean protein by ACHNs, the DH increased from 0 to 180 min, reaching a maximum of 15.58 ± 0.28% after 180 min. In contrast, the DH of DCHNs increased during the first 120 min, after which it changed non-significantly (*p* > 0.05), with a maximum of 7.5 ± 0.45%. The significant difference in the DH of the two hybrid nanoflowers may be due primarily to the specificity of the protease itself. The alcalase in ACHNs is an in-depth endoprotease that efficiently cleaves peptide bonds involving Tyr, Trp, Phe, and Leu and can generate peptides containing hydrophobic and aromatic amino acids. Therefore, it has a broad specificity [38,39]. In contrast, the dispase in DCHNs is a non-specific enzyme that preferentially cleaves the peptide bond between the C-terminal of Leu and Phe at the P1 position [10]. Therefore, ACHNs hydrolyses soybean protein to a greater degree. Wang et al. [40] reported similar results in peony-seed protein hydrolysates and found that the DH of Neutrase hydrolysates (6.19 ± 0.23%) and papain hydrolysates (12.16 ± 0.05%) were lower than that of alcalase (27.97 ± 0.47%) within 0.5 to 4 h of hydrolysis time. The test results of Shuai et al. [3] show that the DH of pea protein treated by two enzymes reached 16.78% (alcalase) and 8.97% (neutrase) at 5 h. This also suggests that the significant difference in the degree of hydrolysis may be mainly due to the specificity of the protease itself.

### 3.3. Structural Characterization of Soybean Protein Hydrolysates

#### 3.3.1. Relative Fluorescence Intensity

Fluorescence intensity and maximum absorption wavelengths of proteins in fluorescence spectra can reflect exposure to aromatic amino acid residues and changes in the polarity of their microenvironment [10]. Figure 4A,B show the fluorescence spectra of natural and hydrolyzed soybean proteins at different time intervals. Compared to the natural soybean protein, the hydrolyzed proteins from hybrid nanoflowers show varying degrees of red-shift and a significant decrease in fluorescence intensity (*p* < 0.05). λmax red-shift indicates that the hydrolysis treatment stretched the peptide chains and exposed the hydrophobic groups embedded within the protein to the aqueous solution. The decrease in fluorescence intensity is mainly attributed to the loss of dense protein structure and the exposure of tryptophan and tyrosine residues to a more polar environment [41]. The increase or decrease in fluorescence intensity is influenced largely by the energy transfer generated by the distances between the two Tyr and Trp and the adjacent fluorescence quenching groups. The reduction in fluorescence intensity observed in this study is likely due either to the reduction in the energy transfer between the two residues or to a rise in the number of fluorescence quenching groups after hydrolysis [40,42].

The fluorescence spectrum of ACHNs-hydrolyzed protein differed from that of DCHNs-hydrolyzed protein. The fluorescence intensity of ACHNs-hydrolyzed protein increased slightly after 120 min of hydrolysis due to the greater DH of ACHNs, which allowed peptides or peptide fragments to aggregate through hydrophobic interactions and thus re-embedded the fluorescent quenching groups inside the aggregates. However, this may also have been due to the conformational changes caused by excessive hydrolysis, resulting in the tryptophan residues being close to the tyrosine residues or away from the fluorescence quenching groups [42]. The fluorescence intensity of ACHNs-hydrolyzed protein increased, as consistent with the findings of this test for DH (Figure 3).

#### 3.3.2. Circular Dichroism

In this study, the changes in the secondary structure content of the SPI after hydrolysis were investigated using CD. Figure 5A,B show that the CD spectrum of SPI was broadly negative and had strong positive peaks at 208 nm and 193 nm, respectively, indicating that the secondary structure of SPI is a highly ordered α-helix and β-sheet [43]. After hydrolysis of the soybean protein using the hybrid nanoflowers, the negative peak blue shifts from 208 nm to 198–202 nm, and its ellipticity is reduced, indicating that the ordered structure of the protein is disrupted, and an α-helix-based ordered conformation changed to a disordered conformation with irregular curls [41,43]. Similar results were observed in the hydrolysis of oat protein isolates using alcalase [44].

Table 1 lists the contents of the secondary structure of SPI and its hydrolysates. Compared to SPI, after hydrolysis by ACHNs and DCHNs, the α-helix and β-sheet contents of the hydrolysates were reduced and the β-turn and random coil contents were increased. However, after 120 min of hydrolysis, the secondary structure of the ACHNs hydrolysates showed that α-helix and β-sheet structures increased while random coil structures decreased. This might be closely related to the electrostatic and conformational changes in the terminal residues and changes in the length of peptides in the hydrolysate [45].

### 3.4. Solubility

The functional properties of proteins are often closely related to their solubility [46]. Figure 6A,B show the changes in solubility of SPI and ACHNs hydrolyzed proteins and DCHNs hydrolyzed proteins at different pH conditions. The solubility of all soybean hydrolysates improved significantly (*p* < 0.05) over the range of pH values evaluated. Fluorescence spectroscopy and CD show that the protein chains unfolded after hydrolysis, exposing more hydrophilic groups and becoming able to adsorb more water molecules, thereby making the protein more hydrophilic and easily soluble in aqueous solutions [41]. At the same time, the tertiary structure of the protein became more porous and facilitated the entry of water molecules via more effective infiltration, swelling, and hydration [9].

In addition, hydrolyzed products with different hydrolysis times had different patterns of variation with pH. The lowest value of the respective solubility was reached at pH 5.0; similar results were reported by Yust et al. [47] in the limited hydrolysis of chickpea protein using alcalase immobilized on glyoxyl-agarose gels. Near the isoelectric point of soybean protein, the molecules of the latter exhibit poor solubility due to reduced intermolecular electrostatic repulsion and this causes severe aggregation of the protein molecules. When SPI is hydrolyzed using hybrid nanoflowers, the tertiary structure of the peptide chain opens, the ratio of hydrophobic and hydrophilic groups on the protein surface changes, and the number of charged groups on the surface increases, improving solubility at the isoelectric point [48,49]. Therefore, the high solubility of hybrid-nanoflower-hydrolyzed SPI at acidic pH values may allow for its use in acidic foods and beverages [50].

The solubility of hydrolysates was also dependent on the type of protease used. The superior solubility of ACHNs-hydrolyzed isolates could be due to their higher DH, which generated a higher number of small peptides. These peptides, which have a low molecular weight, expose more polar residues and have the ability to form hydrogen bonds with water, thus increasing the solubility of the protein.

### 3.5. Emulsifying Properties

The emulsifying properties of protein are measured using the EAI and the ESI. When protein emulsification activity increases, proteins function as emulsifiers with large emulsion interfaces, thereby increasing the specific surface area of droplets and reducing their particle size. The increased protein emulsification activity also helps reduce the settling rate of droplets, which enhances emulsion stability [51,52]. Figure 7A,B show that hydrolysis using hybrid nanoflowers for different reaction times improved the emulsification of SPI significantly (*p* < 0.05). The EAI of ACHNs and DCHNs reached the maximum after hydrolysis for 120 min at 61.38 ± 0.025 m^2^/g and 54.73 ± 0.75 m^2^/g, respectively. The increased emulsification of the proteolytic soybean products may be related to increases in their solubility, peptide chain flexibility, and hydrophobic-group exposure on the surface. However, there was no significant change (*p* > 0.05) in the EAI of DCHNs-hydrolyzed proteins after 180 min, and a decreasing trend was observed in the EAI of ACHNs-hydrolyzed proteins. This phenomenon is due to the high DH: when the peptide chains become shorter and form more hydrophilic peptide segments that are less attractive to the oil-water interface and thus cannot form viscoelastic films at the emulsion oil-drop interface. This leads to a reduction in emulsification [53].

In addition, the emulsion stability of the hydrolysates was significantly higher (*p* < 0.05) than that of the soybean protein isolate at different reaction times. This phenomenon was due mainly to the electrostatic repulsion caused by peptides adsorbed on the surface of the oil droplets [39]. The ESI of DCHNs-hydrolyzed protein reached a maximum value of 50.34 ± 2.2 min after 180 min, while the ESI of ACHNs-digested protein reached a maximum value of 69.96 ± 1.7 min after 120 min. After 120 min, the ESI of ACHNs-hydrolyzed protein decreased significantly (*p* < 0.05). This may have been due to excessive hydrolysis, which increases the difference in density between the organic and water phases. The increase in droplet radius and continuous decrease in phase viscosity results in phase separation [51].

### 3.6. Antioxidant Capacity

DPPH radical-scavenging activity, ABTS radical-scavenging assay, and Fe^2+^ chelating activity are the indicators of the antioxidant capacity of natural products [54,55,56]. As shown in Figure 8, the antioxidant capacity of both hybrid nanoflower hydrolysates was significantly higher than that of SPI (*p* < 0.05). This is because the cleavage sites of the hybrid nanoflowers involve phenylalanine, tyrosine, tryptophan, and lysine carboxyl groups, the phenyl rings of which have powerful electron-donating abilities; this means they can effectively scavenge free radicals and delay oxidation reactions [57]. The results of fluorescence experiments show that exposing tryptophan and tyrosine residues to a polar environment explains this phenomenon.

Furthermore, the antioxidant activity of SPI and hydrolyzed proteins increased with their concentrations, showing a dose-dependence. Furthermore, Figure 8 shows that the antioxidant capacity of the hydrolysate increased with the increase of hydrolysis time at low concentrations. This can be attributed to the fact that as the hydrolysis time increases, more small peptides or amino acids that react with free radicals are released, contributing to the increased antioxidant capacity of the hydrolysate. However, the increase in hydrolysis time of both hybrid nanoflower hydrolysates at high concentrations of 2.5 mg/mL and 250 μg/mL did not have a significant effect on the increase in ABTS radical-scavenging activity and Fe^2+^ chelating ability (*p* > 0.05). This may be due to the fact that the water-soluble ABTS cation solution and Fe^2+^ solution are more readily bound to the hydrolyzed protein compared to the DPPH oil-soluble radicals, resulting in a short hydrolysis time at high concentrations to produce enough hydrolysate to bind almost all of the ABTS cation radicals and Fe^2+^ in solution, resulting in no significant effect of increased hydrolysis time on the improved antioxidant properties of the hydrolysate [12].

As shown in Table 2, the ACHNs hydrolysates all had lower IC_50_ values compared to the DCHNs hydrolysates at the same hydrolysis time. This indicates that the antioxidant activity of ACHNs hydrolysates is stronger due to differences in the peptide-chain lengths and amino acid compositions of hydrolysates obtained using different proteases [58]. The specificity of ACHNs and effectiveness of cleaved peptide bonds produce more amino acids with the ability to scavenge free radicals.

Compared with the antioxidant activity of proteolytic digests reported in the literature, such as the DPPH radical-scavenging activity of whey protein concentrates hydrolyzed with immobilized aspartate protease (IC_50_ = 3.36 mg/mL) [59], ABTS radical-scavenging activity of peony seed hydrolysates hydrolyzed with different proteases (IC_50_ > 1.57 mg/mL)^39^and Fe^2+^ chelating activity of eggshell-membrane protein hydrolysates hydrolyzed with alcalase (IC_50_ > 9.79 mg/mL) [60], the hybrid nanoflower hydrolysates in this study had lower IC_50_ values (Table 2). Therefore, the above results suggest that soybean protein hydrolyzed by hybrid nanoflowers can better stabilize DPPH/ABTS radicals and act as an electron donor to terminate the radical chain reaction. Also, the hybrid nanoflower soybean protein hydrolysate effectively chelates Fe^2+^ as a pro-oxidant, thus delaying the oxidation reaction.

### 3.7. Properties of Soybean Proteolyte Emulsions

These experimental results show that hydrolysis using two different hybrid nanoflowers gives SPI hydrolysates excellent physicochemical properties after a reaction time of 120 min. Therefore, after 120 min of hydrolysis, the hydrolysates were selected to prepare the SPI emulsion in this study. As shown in Table 3 and Figure 9, the particle size and dispersion coefficient of the emulsion prepared using the hybrid-nanoflower-hydrolyzed protein were significantly lower (*p* < 0.05) than of those prepared using natural soybean protein. On the one hand, this may be because of the lower molecular weight of the hydrolyzed protein and the increased flexibility of the protein, which makes diffusion at an oil-water interface easier. On the other hand, it may be because solubility and emulsification are increased after proteolysis, thereby facilitating better adsorption of the hydrolyzed hydrolysis products on the oil-water interface and resulting in a reduced emulsion particle size.

Zeta-potential reflects the surface charge of the emulsion droplets, and both natural and hydrolyzed soybean protein emulsions have a negative zeta-potential. This is because the emulsions were prepared under neutral conditions (where the pH is greater than the isoelectric point (pI) of the protein), resulting in emulsions being negatively charged [61]. Compared to the natural soybean protein emulsion, the absolute value of the zeta-potential of the emulsions stabilized using the protein from the two hybrid nanoflowers increased significantly (*p* < 0.05) due to the increased number of charged groups after hydrolysis. This increased charge on the surface of the droplets strengthened the electrostatic repulsion and could prevent aggregation between droplets during the preparation of emulsions [62].

As Figure 9 shows, the particle size and zeta-potential of soybean protein digests made using ACHNs and DCHNs were different. This is related to the differences in the properties of the two enzymes used. Pan et al. [63] prepared emulsions using rice protein hydrolysates produced using different proteases (neutrase, trypsin, and alcalase) and reported that properties of the emulsions were different and related to the type of protease used.

### 3.8. Oxidation Stability

In this experiment, the oxidative stability of the emulsions made using protein digested using nanoflowers’ enzymes were compared to the nano-emulsions of natural soybean protein. As Figure 10 shows, the emulsions had low POV and TBARS values on the first day of storage. As the storage time increased, the POV and TBARS values for all emulsions increased significantly (*p* < 0.05); however, emulsions stabilized with ACHNs and DCHNs hydrolysates had a significantly lower increase in POV and TBARS values than those stabilized with natural SPI during storage (*p* < 0.05), showing delayed lipid oxidation. The results of Jacobsen et al. [64] showed that this greater oxidative stability increased as the antioxidant properties of the emulsifier and the interfacial charge of the emulsion increased. The hydrolysates protected soybean oil by scavenging free radicals, chelating metal ions, and increasing the thickness of the viscoelastic interfacial layer. This showed the utility of emulsions that effectively inhibit hydrogen release from lipids and lipid peroxidation [65]. On the other hand, it is evident from the potentiometric results that the absolute zeta-potential of the hybrid-nanoflowers-hydrolyzed protein emulsions was significantly higher compared to the natural protein emulsions. The droplets had a relatively high electrostatic repulsive force and were able to bind to the naturally occurring pro-oxidant (Fe^3+^) efficiently in the aqueous phase, improving the oxidative stability of the emulsions [66]. ACHNs hydrolysates had superior functional properties to DCHNs hydrolysates. Furthermore, the absolute zeta-potential of the ACHNs emulsion was higher compared to the DCHNs emulsion, leading to better oxidative stability.

## 4. Conclusions

This study investigated the change in the structure and properties of soybean protein after hydrolysis using ACHNs and DCHNs after different reaction times and the basic properties and oxidative stability of emulsions prepared using soybean protein hydrolysates as emulsifiers. Results show that hydrolysis using hybrid nanoflowers results in a more flexible structure of the soybean protein with excellent functional and antioxidant properties. In addition, the properties of the hydrolysates vary in relation to the protease type and hydrolysis time. When the hydrolysis time is 120 min, the emulsification of ACHNs hydrolysates and DCHNs hydrolysates is the best. When the hydrolysis time is 180 min, ACHNs hydrolysates and DCHNs hydrolysates DPPH radical-scavenging activity, ABTS radical-scavenging activity, and metal ion chelating activity is the best. Under the same DH, ACHNs has the strongest improvement effect on SPI properties, followed by DCHNs. This study also found that the protein hydrolysates can significantly delay the lipid oxidation of oil-in-water emulsions. According to the POV and TBARS values of soybean protein and its hydrolysates nanoemulsion at different storage times, ACHNs hydrolysates improved the antioxidant capacity of the emulsion more than DCHNs hydrolysates. These results strongly support the possible application of hybrid-nanoflowers-hydrolyzed soybean protein as an emulsifier in food (such as drink) and medicinal applications (such as transport of oxidizable active substances). Nonetheless, future studies are still needed to demonstrate the potential of hydrolyzed protein-stabilized emulsions of hybrid nanoflowers for the delivery of bioactive substances, as well as to explore the storage stability, oxidative stability, and digestive stability of this delivery system in order to provide more theoretical support for the practical application of hybrid nanoflowers.

## Figures and Tables

**Figure 1 foods-11-03409-f001:**
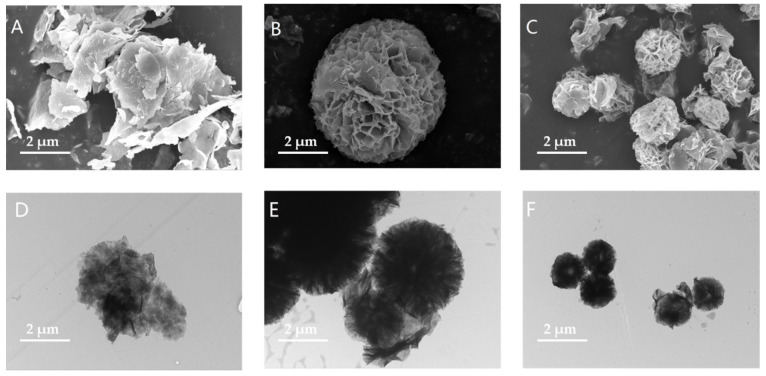
SEM images of (**A**) Cu_3_(PO_4_)_2_·3H_2_O; (**B**) ACHNs; and (**C**) DCHNs. TEM images of (**D**) Cu_3_(PO_4_)_2_·3H_2_O; (**E**) ACHNs; and (**F**) DCHNs.

**Figure 2 foods-11-03409-f002:**
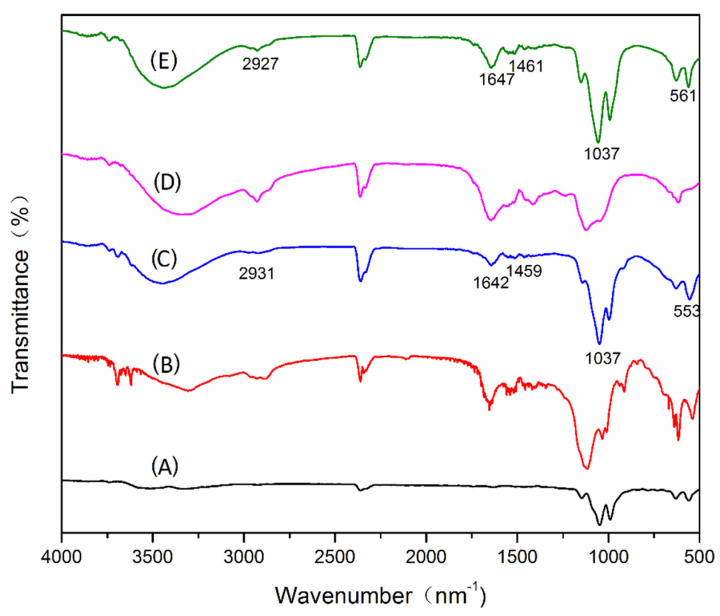
FT−IR spectra of (**A**) Cu_3_(PO_4_)_2_·3H_2_O; (**B**) free alcalase; (**C**) ACHNs; (**D**) free dispase; and (**E**) DCHNs.

**Figure 3 foods-11-03409-f003:**
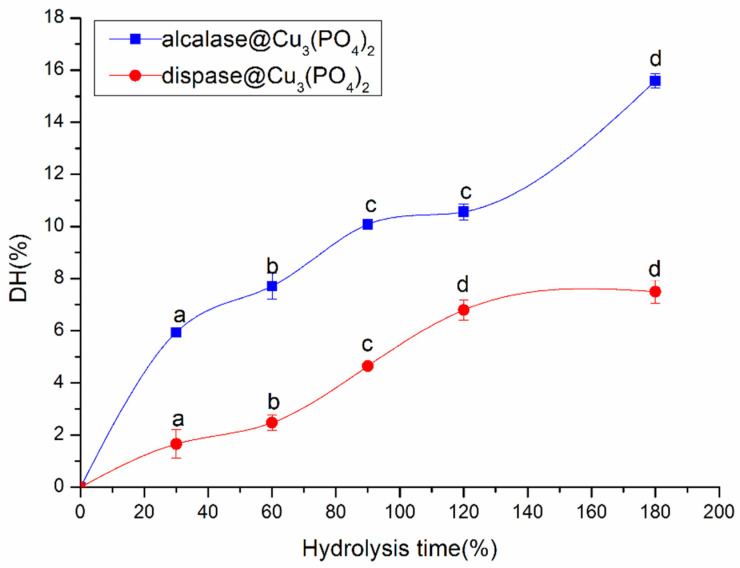
Hydrolysis curves of soybean protein hydrolyzed by ACHNs and DCHNs. Different letters (a–d) represent significant differences between the hydrolysis degrees obtained for different hydrolysis times of the same protease (*p* < 0.05).

**Figure 4 foods-11-03409-f004:**
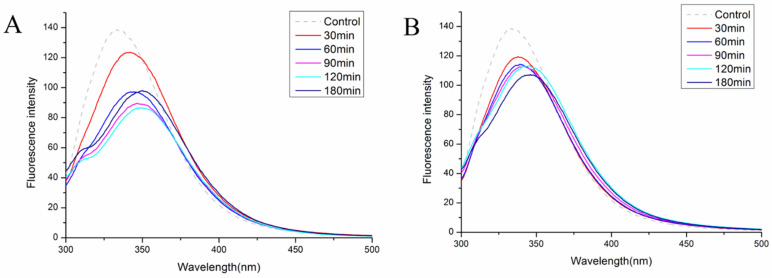
Structural characterization of SPI and hydrolysates. Relative fluorescence spectra of SPI and hydrolysates prepared with ACHNs (**A**) and DCHNs (**B**).

**Figure 5 foods-11-03409-f005:**
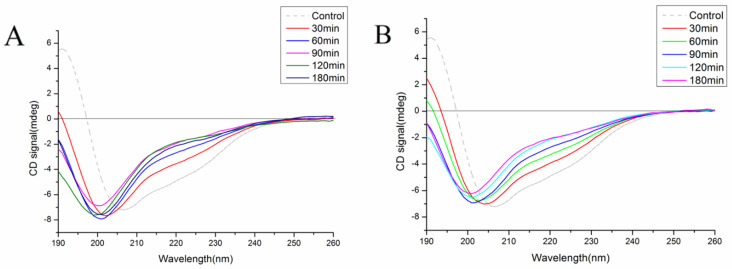
Circular dichroism (CD) spectra of SPI and hydrolysates prepared with ACHNs (**A**) and DCHNs (**B**).

**Figure 6 foods-11-03409-f006:**
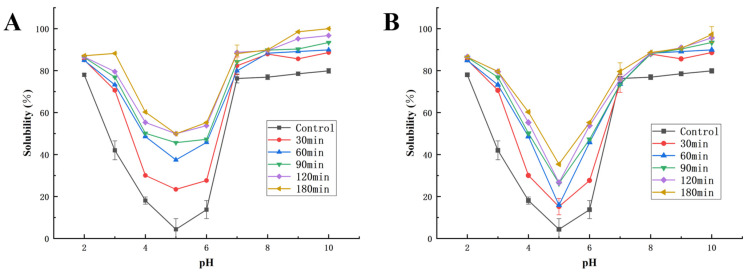
Solubility at different pH: (**A**) Solubility of ACHNs; (**B**) Solubility of DCHNs.

**Figure 7 foods-11-03409-f007:**
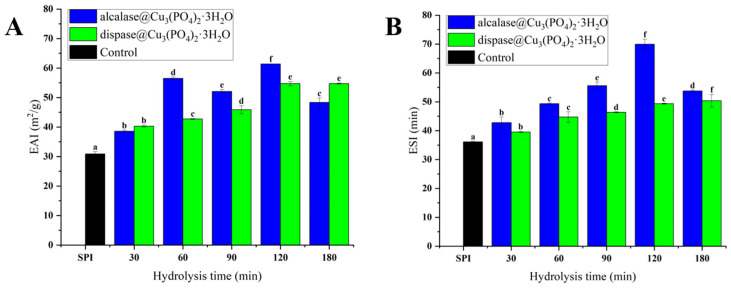
ACHNs, and DCHNs at different hydrolysis times: (**A**) EAI and (**B**) ESI. Different letters (a–f) represent significant differences between hydrolysates produced by the same enzyme hydrolysis at different times(*p* < 0.05).

**Figure 8 foods-11-03409-f008:**
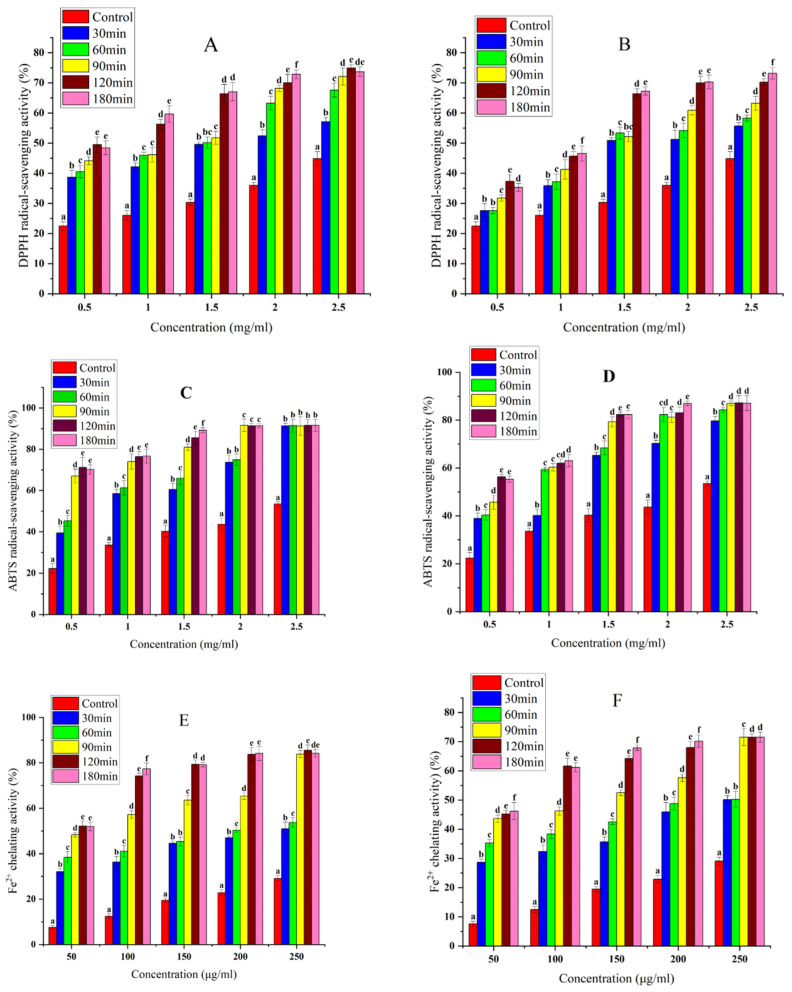
Radical-scavenging activity of SPI and its hydrolysates at different concentrations. DPPH radical-scavenging activity: (**A**) ACHNs; (**B**) DCHNs, ABTS radical-scavenging activity; (**C**) ACHNs; (**D**) DCHNs, metal ion chelating activity: (**E**) ACHNs; (**F**) DCHNs. Different letters (a–f) indicate significant differences among hydrolysates at time function for the same concentration(*p* < 0.05).

**Figure 9 foods-11-03409-f009:**
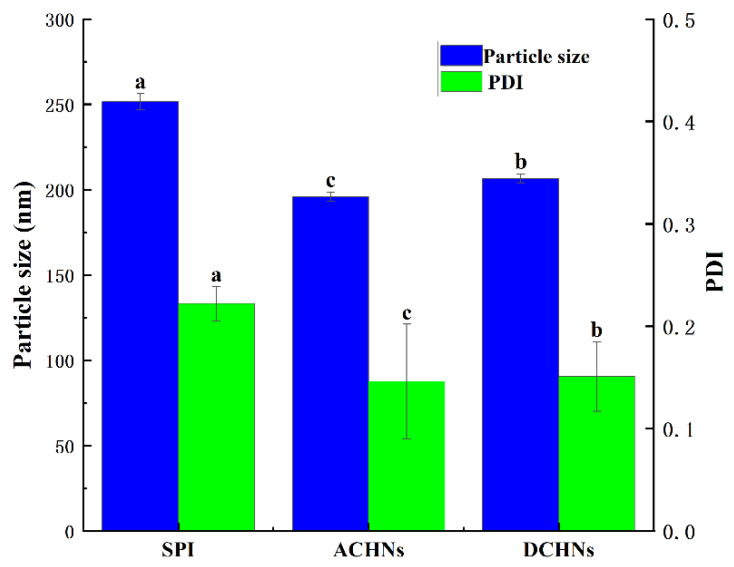
Particle size of modified soybean protein emulsion. Different letters in the same column indicate a significant difference (*p* <0.05).

**Figure 10 foods-11-03409-f010:**
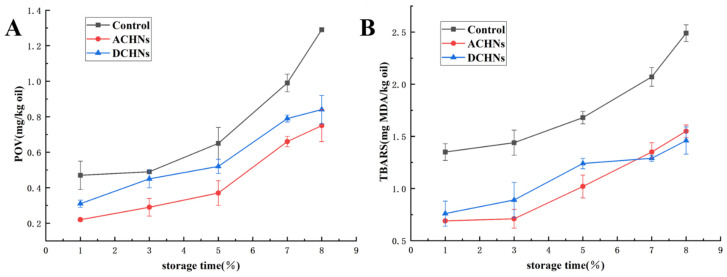
(**A**) POV and (**B**) TBARS values of soybean protein and its hydrolysates nanoemulsion at different storage times.

**Table 1 foods-11-03409-t001:** Secondary structure content of SPI (soybean protein) and hydrolysates.

Sample	α-Helical (%)	β-Sheet (%)	β-Turn (%)	Random Structure (%)
SPI	11.1	43.0	19.3	26.6
ACHNs 30 min	7.5	38.1	21.3	33.1
ACHNs 60 min	6.9	36.7	21.6	34.7
ACHNs 90 min	6.4	37.8	20.9	34.8
ACHNs 120 min	6.0	34.6	21.3	38.3
ACHNs 180 min	6.6	36.2	21.9	35.9
SPI	11.1	43.0	19.3	26.6
DCHNs 30 min	8.8	42.2	21.0	31.4
DCHNs 60 min	7.9	41.0	20.9	32.3
DCHNs 90 min	7.1	38.9	21.1	33.6
DCHNs 120 min	6.7	38.8	20.8	34.0
DCHNs 180 min	6.6	38.4	21.4	34.1

**Table 2 foods-11-03409-t002:** IC_50_ values of the antioxidant activities of SPI and SPHs.

Sample	IC_50_ Values of DPPH (mg/mL)	IC_50_ Values of ABTS (mg/mL)	IC_50_ Values of Fe^2+^ Chelating (μg/mL)
	ACHNs DCHNs	ACHNs DCHNs	ACHNs DCHNs
SPI	4.589 ± 0.318 ^A^	2.385 ± 0.101 ^A^	625.24 ± 10.365 ^A^
30 min	1.557 ± 0.089 ^F,b^ 1.787 ± 0.057 ^E,a^	0.776 ± 0.052 ^F,b^ 0.919 ± 0.052 ^F,a^	246.301 ± 6.159 ^F,b^ 296.928 ± 5.364 ^F,a^
60 min	1.058 ± 0.044 ^E,b^ 1.590 ± 0.053 ^D,a^	0.655 ± 0.025 ^E,b^ 0.715 ± 0.059 ^E,a^	201.895 ± 5.121 ^E,b^ 262.719 ± 5.970 ^E,a^
90 min	0.908 ± 0.058 ^D,b^ 1.321 ± 0.026 ^C,a^	0.284 ± 0.005 ^D,b^ 0.610 ± 0.046 ^D,a^	63.149± 3.926 ^D,b^ 98.276 ± 2.982 ^D,a^
120 min	0.571 ± 0.013 ^C,b^ 0.927 ± 0.061 ^B,a^	0.219 ± 0.019 ^B,b^ 0.450 ± 0.011 ^C,a^	43.210 ± 2.458 ^C,b^ 60.352 ±1.597 ^C,a^
180 min	0.553 ± 0.009 ^B,b^ 0.930 ± 0.033 ^B,a^	0.239 ± 0.010 ^C,b^ 0.467 ± 0.017 ^B,a^	40.947 ± 3.685 ^B,b^ 57.024 ± 2.636 ^B,a^

Note: Different letters (a,b) represent significant differences between the hydrolysates produced during the same hydrolysis times for different proteases (*p* < 0.05). Different letters (A–F) represent significant differences between the hydrolysates produced during different hydrolysis times for the same protease (*p* < 0.05).

**Table 3 foods-11-03409-t003:** Particle size and zeta-potential of modified soybean protein emulsion.

Sample	Particle Size (nm)	PDI	Zeta-Potential (mV)
SPI	251.6 ± 4.666 ^a^	0.222 ± 0.017 ^a^	−37.2 ± 0.76 ^c^
ACHNs	195.8 ± 2.757 ^c^	0.146 ± 0.056 ^c^	−45.5 ± 0.61 ^a^
DCHNs	206.7 ± 2.598 ^b^	0.151 ± 0.034 ^b^	−43.7 ± 0.62 ^b^

Note: Different letters in the same column indicate a significant difference (*p* < 0.05).

## Data Availability

All related data and methods are presented in this paper. Additional inquiries should be addressed to the corresponding author.

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
