# Peer review of "Physicochemical Antioxidative and Emulsifying Properties of Soybean Protein Hydrolysates Obtained with Dissimilar Hybrid Nanoflowers"

_foods, 2022, doi:10.3390/foods11213409_

Round 1
Reviewer 1 Report
This work present Physicochemical and Antioxidative Properties of Soybean Protein Hydrolysates Obtained with Dissimilar Hybrid Nanoflowers. This manuscript (MS) need changes which are mentioned as following
Paper format: The paper is in foods required format.
Title: The MS title sound good, however, need to change to change the title. Its better to keep emulsifying properties of soybean protein hydrolysates
Abstract: The abstract should always be concise and informative. The arguments of why your study is important not making any sense. Extensive revision is required in abstract, as the present sentences sounds noisy during reading. Overall the abstract is not informative enough and need to show the actual picture of the work. Authors are supposed to please indicate the numerical values with significant difference in abstract. Keep abstract minimum with covering all of your data.
Keywords: Not according to manuscript. Please change it to some structural features study
Introduction: some shortcoming are below
1. Why this modification method have advantage over others methods?
2. Add more of information why other methods are not satisfactory for modification?
3. Its always concern with final residue with enzyme treatment, what was the expected residue in final product?
4. Novelty is missing in introduction, add few sentence to justify your work, with proper aim and objectives.
5. What is the safety and environment concern for such methods? elaborate
6. Please add/indicate and compare your work with pervious published paper.
7. Please cite the following latest papers for oil-in-water emulsion for modified protein
https://www.tandfonline.com/doi/full/10.1080/10408398.2022.2081665
https://www.sciencedirect.com/science/article/pii/S0268005X1630220X
https://link.springer.com/article/10.1007/s10068-019-00590-z
https://www.mdpi.com/1420-3049/27/11/3646
https://www.ncbi.nlm.nih.gov/pmc/articles/PMC9368430/
Methods and materials:
1. These experiments should be divide in three parts i.e. structural study of hydrolysate, physiochemical properties and then functional properties. Already suggest to change the title as the paper composition is different from the title. Go with three parts.
2. Why characterization of nanofalower was carried out? Is it not in pure form? Why market nanoflowers use in this study was avoided? Same with soybean protein isolate, no need to extract in fresh from the source. If extracted, what was the purity of that protein isolates?
3. Change 2.9 to just functional properties. And yes solubility came under physicochemical study, put it in physicochemical section, as suggested in point no 1.
4. What was the working head of the homogenizer in 2.9.2? Mention please.
5. Up to my understanding the 2 minutes homogenization with that much higher speed can demolish the protein structure. I think it’s not fair, just one minute was not enough? Please explain here why such long time was used here?
6. Mention dilution value (N) in equation no 4.
7. For any emulsion study, viscosity is key point to determine the functional properties, no such experiment was done in this study. Strange
8. What are the importance of zeta potential in emulsion study?
9. No flocculation or any Coalescence were reported in emulsion. Better to describe these important parameters in this study.
Results and discussion:
1. I have through the results and discussion, discussion is fair enough but figures in this study are very poor, hardly to understand the point which authors are trying to address. For example see SEM photos, I did not understand what authors are looking in these photos. All of the figures are unreadable.
2. Please provide curve for particle size experiments, tabulation is not enough.
3. IC50 values should present with antioxidant results? Why it presented separately? Change it please.
4. Over all the figure are too poor to understand, provide readable version of all figures, especially SEM, SEM study is poorly presented with poor photos.
Conclusion: Change the conclusion as per changes in discussion, add more of application for this research for common reader.
Author Response
Comments and Suggestions for Authors:
This work present Physicochemical and Antioxidative Properties of Soybean Protein Hydrolysates Obtained with Dissimilar Hybrid Nanoflowers. This manuscript (MS) need changes which are mentioned as following:
Answer: Thank you very much for taking time to review our manuscript and thanks for your kindly suggestions and detailed comments, which significantly improve our manuscript. We have revised the manuscript according to your kind advices and detailed suggestions.
Comment 1: Paper format: The paper is in foods required format.
Answer: Thank you very much for taking time to review our manuscript. According to the comment, we have modified the format.
Comment 2: Title: The MS title sound good, however, need to change to change the title. Its better to keep emulsifying properties of soybean protein hydrolysates.
Answer: Thanks for your comment. We have corrected the Title.
Title: Physicochemical,Antioxidative and Emulsifying Properties of Soybean Protein Hydrolysates Obtained with Dissimilar Hybrid Nanoflowers
Comment 3: Abstract: The abstract should always be concise and informative. The arguments of why your study is important not making any sense. Extensive revision is required in abstract, as the present sentences sounds noisy during reading. Overall the abstract is not informative enough and need to show the actual picture of the work. Authors are supposed to please indicate the numerical values with significant difference in abstract. Keep abstract minimum with covering all of your data.
Answer: Thank you very much for your careful reading. I’m sorry for our negligence. It was corrected. We have corrected the Abstract,indicate the numerical values with significant difference in abstract.
Abstract (Line 24-25 (the latest version of the manuscript)): The structure and functional properties of soybean protein treated with hybrid nanoflowers were then characterized.
Abstract (Line 28-31 (the latest version of the manuscript)): The solubility of the hydrolysates was significantly higher (p < 0.05) than the soybean protein (SPI) at all pH values tested (2.0 - 10.0), at the same pH value, the solubility of ACHNs hydrolysates and DCHNs hydrolysates the highest increased by 46.2% and 42.2%, respectively.
Abstract (Line 33-36 (the latest version of the manuscript)): The emulsifying activity index of ACHNs hydrolysates and DCHNs hydrolysates reached the maximum after hydrolysis for 120 min at 61.38 ± 0.025 m2/g and 54.73 ± 0.75 m2/g, respectively. It was concluded that the two hydrolysates had better solubility and antioxidant properties, which provided a theoretical basis for SPI product development.
Comment 4: Keywords: Not according to manuscript. Please change it to some structural features study.
Answer: Thanks for your comment. We have corrected the Keywords.
Keywords (Line 40-41 (the latest version of the manuscript)): Hybrid nanoflowers; Soy protein; Structure properties; Functional properties; Emulsion
Comment 5: Introduction: some shortcoming are below.
Answer: Thanks for your comment and I'm sorry for my poor handling of the details of the Introduction.
1.Why this modification method have advantage over others methods?
Answer: Thank you very much for taking time to review our manuscript. According to the comments, the sentences were rewritten.
Line 57-61 (the latest version of the manuscript): Enzymatic modification of proteins is a protein modification means to convert proteins into peptides of different molecular weights by using proteases to catalyze the hydrolysis of peptide bonds in proteins under suitable conditions. Compared with physical and chemical modifications, enzymatic modification has the advantages of less by-products, high specificity and easy control, and is widely used. [3]
2.Add more of information why other methods are not satisfactory for modification?
Answer: Thank you very much for your careful reading. I’m sorry for our negligence. It was corrected.
Line 51-57 (the latest version of the manuscript): But, physical modification has no addition of exogenous substances, but the modification effect is limited; chemical modification is the induction of new organic groups into the protein molecule, which causes changes in the protein main chain or side chains, resulting in changes in the protein structure and physicochemical properties. However, chemical modification also has limitations in its application in the food industry because of the addition of chemicals that may be harmful to human health.
3.Its always concern with final residue with enzyme treatment, what was the expected residue in final product?
Answer: Thank you very much for your questions about this paper. In our paper, the exposure of amino acid residues in the end products of enzymatic hydrolysis was determined by different hybrid nanoflower and hydrolysis time. The purpose of our experiment is to determine the protein structure according to the intensity and red or blue shift of the fluorescence spectrum of the final product of enzymatic hydrolysis, and to obtain the situation of protein hydrolysis by different hybrid nanoflower through structure analysis. At the same time, the degree of protein hydrolysis of the hybrid nanoflower was indicated by the homogeneity of the residual amino acid of the protein. Therefore, the structure of the end product is not predicted in advance.
4.Novelty is missing in introduction, add few sentence to justify your work, with proper aim and objectives.
Answer: Thanks for your comment. It was corrected.
Line 97-99 (the latest version of the manuscript): This experiment is the first time to compare the structural and functional properties of the hydrolysates of nanoflowers hybrid with alkaline protease and neutral protease.
Line 103-104 (the latest version of the manuscript): Thus, it is helpful to explore the application potential and promote the research and development of healthy functional food.
Line 106-109 (the latest version of the manuscript): To explore the use of protein hydrolysates as potential functional components, this laid a theoretical foundation for the development of protein hydrolysates obtained using hybrid nanoflowers with the excellent performance required for application in the medicinal and food industries.
5.What is the safety and environment concern for such methods? Elaborate
Answer: Thank you very much for your comments. We have reviewed the references before the experiment and found that this method is highly safe and will not cause environmental pollution. The amount of reagents we use is within the safe limits.
6.Please add/indicate and compare your work with pervious published paper.
Answer: Thank you very much for your careful reading and detailed comment.
Line 73-75 (the latest version of the manuscript): In addition, the experimental results of Xu et al. [10] showed that the degree of protein hydrolysis also had an effect on the stability of the emulsions prepared from the hydrolysates.
7.Please cite the following latest papers for oil-in-water emulsion for modified protein.
(1)https://www.tandfonline.com/doi/full/10.1080/10408398.2022.2081665
(2)https://www.sciencedirect.com/science/article/pii/S0268005X1630220X
(3)https://link.springer.com/article/10.1007/s10068-019-00590-z
(4)https://www.mdpi.com/1420-3049/27/11/3646
(5)https://www.ncbi.nlm.nih.gov/pmc/articles/PMC9368430/
Answer: Thanks for your comment. I have cited the above five documents in the following places.
- as a reference in the latest paper [14], in Line 84 (the latest version of the manuscript).
(2) as a reference in the latest paper [10], in Line 73、358 and 373 (the latest version of the manuscript).
(3) as a reference in the latest paper [20], in Line 144 (the latest version of the manuscript).
(4) as a reference in the latest paper [29], in Line 217 (the latest version of the manuscript).
(5) as a reference in the latest paper [3], in Line 61 and 361 (the latest version of the manuscript).
Comment 6: Methods and materials:
1.These experiments should be divide in three parts i.e. structural study of hydrolysate, physiochemical properties and then functional properties. Already suggest to change the title as the paper composition is different from the title. Go with three parts.
Answer: Thanks for your comment. I have changed this part.
Line 205 (the latest version of the manuscript): 2.9. Physiochemical Properties of SPI and Its hydrolysates
Line 206 (the latest version of the manuscript): 2.9.1. Solubility
Line 215 (the latest version of the manuscript): 2.10. Functional Properties of SPI and Its hydrolysates
Line 216 (the latest version of the manuscript): 2.10.1. Emulsifying Properties
Line 231 (the latest version of the manuscript): 2.10.2. Antioxidant Capacity
Line 232 (the latest version of the manuscript): 2.10.2.1. DPPH Radical Scavenging Activity
Line 244 (the latest version of the manuscript): 2.10.2.2. ABTS Radical Scavenging Activity
Line 258 (the latest version of the manuscript): 2.10.2.3. Metal Ion Chelating Activity
Line 268 (the latest version of the manuscript): 2.10.2.4. Determination of IC50 Values
Line 272 (the latest version of the manuscript): 2.11. Preparation of Protein Emulsions
Line 283 (the latest version of the manuscript): 2.12. Characterizing the Emulsion
Line 284 (the latest version of the manuscript): 2.12.1. Particle Size and Zeta (ζ) Potential Measurement
Line 289 (the latest version of the manuscript): 2.12.2. Oxidation Stability
Line 307 (the latest version of the manuscript):2.13. Statistical Analysis
2.Why characterization of nanofalower was carried out? Is it not in pure form? Why market nanoflowers use in this study was avoided? Same with soybean protein isolate, no need to extract in fresh from the source. If extracted, what was the purity of that protein isolates?
Answer: Thanks for your comment. I explained it as follows.
(1) The reason we're characterizing nanoflower is because we want to verify that nanoflower is formed.
(2) Nanoflower is commercially available, but the nanoflower used in this article has not been reported.
(3) The preparation process of soybean protein isolate purchased on the market is not very clear, which may affect the properties of soybean protein. The soybean protein prepared by our laboratory has little influence on the structure of the protein, which is conducive to the study of the properties of the later experiment. Therefore, self-made soybean protein was selected for the experiment. The content of soy protein is 97%.
3.Change 2.9 to just functional properties. And yes solubility came under physicochemical study, put it in physicochemical section, as suggested in point no 1.
Answer: Thanks for your comment. It has been modified according to 1.
4.What was the working head of the homogenizer in 2.9.2? Mention please.
Answer: Thank you very much for your careful reading. The homogenizer is FJ200-SH (Shanghai Huxi Industry Co., Ltd, Shanghai, China), the working head specification is Φ12mm/Φ18mm
5.Up to my understanding the 2 minutes homogenization with that much higher speed can demolish the protein structure. I think it’s not fair, just one minute was not enough? Please explain here why such long time was used here?
Answer: Thank you very much for your careful reading and comments. This rotational speed was based on our previous research. Specific reference is as follows:
Emulsification and bacteriostasis of esterified soybean protein - chitosan complex DOI: 10.13550/j.jxhg.20220055
Methods and materials 1.2.6: 12 mL of esterified soybean protein, esterified soybean protein (pH 3.0-5.0) and esterified soybean protein-chitosan complex solutions with a mass concentration of 10 g/L were taken, and 4mL corn oil was added for homogenization (homogenization at 10000 r/min for 2 min) to prepare emulsion with a volume fraction of 25% oil phase. At 0 and 10 min, 40 μL of emulsion was taken from the bottom and mixed with 5 mL of 1 g/L SDS solution. After shaking, the absorbance was measured at 500 nm. The emulsifying activity (EAI, m2/g) and emulsifying stability (ESI, min) of the samples were calculated according to the following equation.
People in our laboratory have been using this machine to make emulsions without protein denaturation under the same conditions. The emulsion is also made at similar speed and time in the literature, and the experimental results of other personnel in our laboratory are very correct after comparison with the results of the literature. The results of preliminary experiments showed that the effect of 2 minutes was better than that of 1 minute, and the phenomenon of protein denaturation did not occur.
6.Mention dilution value (N) in equation no 4.
Answer: Thanks for your comment.
Line 229 (the latest version of the manuscript): where N is the dilution factor (125), φ refers to the oil fraction (0.25).
7.For any emulsion study, viscosity is key point to determine the functional properties, no such experiment was done in this study. Strange
Answer: First of all, thank you for your question. I feel very sorry for not characterizing the viscosity, because this experiment mainly focuses on the structural properties, physical and chemical properties and functional properties of the enzymatic hydrolysate. The emulsion is an extension of emulsification, which verifies that the antioxidant resistance of the emulsion will be improved when the enzymatic hydrolysate is used as raw material. We will take the viscosity into account if we expand the emulsion in the future.
8.What are the importance of zeta potential in emulsion study?
Answer: Thank you very much for your careful reading. Your serious attitude towards your work is very worthy of my learning. Zeta potential is a very important index for emulsion. The higher its absolute value, the better its stability.
9.No flocculation or any Coalescence were reported in emulsion. Better to describe these important parameters in this study.
Answer: Thanks for your comment. In the study, the particle size, PDI and oxidation stability of the emulsion were measured, which indicated the stability of the emulsion to a certain extent. The emulsion did not appear flocculation or any Coalescence. This phenomenon also did not appear in our previous experiments.
Comment 7: Results and discussion:
1.I have through the results and discussion, discussion is fair enough but figures in this study are very poor, hardly to understand the point which authors are trying to address. For example see SEM photos, I did not understand what authors are looking in these photos. All of the figures are unreadable.
Answer: Thanks for your comment. According to your suggestions, I updated the ruler,SEM photos have been modified. Through SEM images with the same magnification, we can see the appearance of nanoflowers and judge whether nanoflowers are formed. New drawings have been replaced in the manuscript and attachments.
2.Please provide curve for particle size experiments, tabulation is not enough.
Answer: Thanks for your comment. We have modified the problem.
Line 547-548 (the latest version of the manuscript): Figure 9. Particle size of modified soybean protein emulsion. Different letters in the same column indicate a significant difference (p <0.05).
3.IC50 values should present with antioxidant results? Why it presented separately? Change it please.
Answer: Thanks for your comment. we have modified it,IC50 Values have been incorporated into the Oxidation Stability.
4.Over all the figure are too poor to understand, provide readable version of all figures, especially SEM, SEM study is poorly presented with poor photos.
Answer: Thank you very much for taking time to review our manuscript. According to the comment, we reworked all the drawings.
Comment 8: Conclusion: Change the conclusion as per changes in discussion, add more of application for this research for common reader.
Answer: Thanks for your comment. I apologize again for the lack of clarity in my language.
Line 591-613 (the latest version of the manuscript): This study investigated the change in the structure and properties of soybean protein after hydrolysis using ACHNs and DCHNs after different reaction times and the basic properties and oxidative stability of emulsions prepared using soybean protein hydrolysates as emulsifiers. Results showed that hydrolysis using hybrid nanoflowers resulted in a more flexible structure of the soybean protein with excellent functional and antioxidant properties. In addition, the properties of the hydrolysates varied in relation to the protease type and hydrolysis time. When the hydrolysis time was 120 min, the emulsification of ACHNs hydrolysates and DCHNs hydrolysates was the best. When the hydrolysis time is 180min, ACHNs hydrolysates and DCHNs hydrolysates DPPH radical scavenging activity, ABTS radical scavenging activity and metal ion chelating activity is the best. Under the same DH, ACHNs had the strongest improvement effect on SPI properties, followed by DCHNs. This study also found that the protein hydrolysates can significantly delay the lipid oxidation of oil-in-water emulsions. According to the POV and TBARS values of soybean protein and its hydrolysates nanoemulsion at different storage time, ACHNs hydrolysates improved the antioxidant capacity of the emulsion more than DCHNs hydrolysates. These results strongly support the possible application of hybrid nanoflowers-hydrolyzed soybean protein as an emulsifier in the food (such as drink) and medicinal applications (such as transport of oxidizable active substances). Nonetheless, future studies are still needed to demonstrate the potential of hydrolyzed protein-stabilized emulsions of hybrid nanoflowers for the delivery of bioactive substances, as well as to explore the storage stability, oxidative stability and digestive stability of this delivery system to provide more theoretical support for the practical application of hybrid nanoflowers.
Reviewer 2 Report
I reviewed the manuscript entitled, Physicochemical and Antioxidative Properties of Soybean Protein Hydrolysates Obtained with Dissimilar Hybrid Nanoflowers. The manuscript is well written and scientifically sounds high. Research hypothesis and approach is novel and contributes to the field.
Abstract:
Lines 33: how much improved
Before line 48: introduce about hydrolysis
Introduction
Well written and appropriate
Line 93: Supported,……….. could you change the word?
Line 110: done…can be replaced by performed
Section 2.3: provide the details of each method
Line 140: Practice.. can be replaced with method
Line 150: provide space before to
Line 154: please revise “Equation 2 describes how the DH values were calculated”
Methodology section is appropriate and cited the ref in methodology
Results and discussion
Figures 2 -5: quality must be improved, x AND Y AXIS
3.2. Degrees of Hydrolysis: This section should be discussed with more available literature
Conclusions should be revised
References are not according to the journal format. Please revise carefully
Author Response
Comments and Suggestions for Authors:
I reviewed the manuscript entitled, Physicochemical and Antioxidative Properties of Soybean Protein Hydrolysates Obtained with Dissimilar Hybrid Nanoflowers. The manuscript is well written and scientifically sounds high. Research hypothesis and approach is novel and contributes to the field.
Answer: Thank you very much for taking time to review our manuscript and thanks for your kindly suggestions and detailed comments, which significantly improve our manuscript. We have revised the manuscript according to your kind advices and detailed suggestions and improved the quality of language.
Comment 1: Lines 33: how much improved.
Answer: Thanks for your comment. We have corrected the Line33.
Line 28-31 (the latest version of the manuscript): The solubility of the hydrolysates was significantly higher (p < 0.05) than the soybean protein (SPI) at all pH values tested (2.0 - 10.0), at the same pH value, the solubility of ACHNs and DCHNs the highest increased by 46.2% and 42.2%, respectively.
Comment 2: Before line 48: introduce about hydrolysis.
Answer: Thank you very much for your careful reading. I’m sorry for our negligence. It was corrected.
Line 51-61 (the latest version of the manuscript): But, physical modification has no addition of exogenous substances, but the modification effect is limited; chemical modification is the induction of new organic groups into the protein molecule, which causes changes in the protein main chain or side chains, resulting in changes in the protein structure and physicochemical properties. However, chemical modification also has limitations in its application in the food industry be-cause of the addition of chemicals that may be harmful to human health. Enzymatic modification of proteins is a protein modification means to convert proteins into pep-tides of different molecular weights by using proteases to catalyze the hydrolysis of peptide bonds in proteins under suitable conditions. Compared with physical and chemical modifications, enzymatic modification has the advantages of less by-products, high specificity and easy control, and is widely used. [3]
Comment 3: Line 93: Supported,……….. could you change the word?
Answer: Thank you very much for your careful reading and comments.
Line 113-114 (the latest version of the manuscript): Alcalase was purchased from Novozymes Biotechnology Co., Ltd. (Tianjin, China).
Comment 4: Line 110: done…can be replaced by performed.
Answer: Thanks for your comment and I'm sorry for my poor handling of the details of the article. Now it has been modified together with Comment 5.
Comment 5: Section 2.3: provide the details of each method.
Answer: Thank you very much for your advice. The method of 2.3 has been described in detail.
Line 130-144 (the latest version of the manuscript): The nanoflower lyophilized samples were gold sprayed to coat the surface with gold. Then, observation was performed using a scanning electron microscope (SEM) (SU8010, Hitachi, Japan) at an accelerated voltage of 5.0 KV.
The microstructure of the nanoflower samples was observed using H-7650 trans-mission electron microscope (TEM) (H-7650, Hitachi, Japan). The nanoflowers were diluted to a certain multiple and the diluted nanoflower samples were dropped onto a carbon-coated copper grid and adsorbed for 1 h. Then, the samples were stained with 2 % (w/v) phosphotungstic acid negative staining solution for 2 min. After the samples were air-dried at room temperature, the nanoflower samples were observed at an accelerating voltage of 100 kV.
In conjunction with Fourier transform infrared (FTIR) spectroscopy (Bruker Ver-tex 70, Bruker Optics, Ettlingen, Germany). The nanoflower samples were mixed with potassium bromide (1:100, w/w) and pressed into tablets for processing. The infrared spectra were scanned in the wavelength range of 4000-400 cm-1 at a resolution of 4 cm-1 and a scanning frequency of 64 times [20].
Comment 6: Line 140: Practice.. can be replaced with method.
Answer: Thanks for your comment. It was corrected.
Line 160-161 (the latest version of the manuscript): The SPI was extracted using the practice reported by Zheng et al. [22] with some modifications.
Comment 7: Line 150: provide space before to.
Answer: Thank you very much for your careful reading and detailed comment. We have modified Line 181 (the latest version of the manuscript).
Comment 8: Line 154: please revise “Equation 2 describes how the DH values were calculated”.
Answer: Thanks for your comment. I apologize for not describing the sentence clearly. We have reformulated Line 185(the latest version of the manuscript).
Line 185 (the latest version of the manuscript): ‘‘The DH was determined using the following formula:’’
Comment 9: Figures 2 -5: quality must be improved, x AND Y AXIS
Answer: Thanks for your comment. We have modified the Figures 2 -5. The new drawings have been incorporated into the new manuscript.
Comment 10: 3.2. Degrees of Hydrolysis: This section should be discussed with more available literature
Answer: Thanks for your comment. We have rephrased 3.2.
Line 361-364 (the latest version of the manuscript): Shuai et al. [3]the test results show that the DH of pea protein treated by two enzymes reached 16.78% (alcalase), 8.97% (neutrase) at 5 h. This also suggests that the significant difference in the degree of hydrolysis may be mainly due to the specificity of the protease itself.
Comment 11: Conclusions should be revised.
Answer: Thanks for your comment. We have modified the sentences.
Line 591-613 (the latest version of the manuscript): This study investigated the change in the structure and properties of soybean protein after hydrolysis using ACHNs and DCHNs after different reaction times and the basic properties and oxidative stability of emulsions prepared using soybean protein hydrolysates as emulsifiers. Results showed that hydrolysis using hybrid nanoflowers resulted in a more flexible structure of the soybean protein with excellent functional and antioxidant properties. In addition, the properties of the hydrolysates varied in relation to the protease type and hydrolysis time. When the hydrolysis time was 120 min, the emulsification of ACHNs hydrolysates and DCHNs hydrolysates was the best. When the hydrolysis time is 180min, ACHNs hydrolysates and DCHNs hydrolysates DPPH radical scavenging activity, ABTS radical scavenging activity and metal ion chelating activity is the best. Under the same DH, ACHNs had the strongest improvement effect on SPI properties, followed by DCHNs. This study also found that the protein hydrolysates can significantly delay the lipid oxidation of oil-in-water emulsions. According to the POV and TBARS values of soybean protein and its hydrolysates nanoemulsion at different storage time, ACHNs hydrolysates improved the antioxidant capacity of the emulsion more than DCHNs hydrolysates. These results strongly support the possible application of hybrid nanoflowers-hydrolyzed soybean protein as an emulsifier in the food (such as drink) and medicinal applications (such as transport of oxidizable active substances). Nonetheless, future studies are still needed to demonstrate the potential of hydrolyzed protein-stabilized emulsions of hybrid nanoflowers for the delivery of bioactive substances, as well as to explore the storage stability, oxidative stability and digestive stability of this delivery system to provide more theoretical support for the practical application of hybrid nanoflowers.
Comment 12: References are not according to the journal format. Please revise carefully.
Answer: Thanks for your comment. According to your suggestions, we have updated the references.

Round 2
Reviewer 1 Report
Improved